# Inverse Tangent Functional Nonlinear Feedback Control and Its Application to Water Tank Level Control

**Jian Zhao** and **Xianku Zhang** *

Navigation College, Dalian Maritime University, Dalian 116026, China; zhaojian@dlmu.edu.cn
* Correspondence: zhangxk@dlmu.edu.cn

**Abstract:** This paper explores the significance and feasibility of addressing a notion that the system error of a nonlinear feedback control can be decorated by an inverse tangent function in order to attain a sound energy-efficient performance. The related mathematical model and relevant evaluation of this concept are further illustrated by demonstrating a case study about the control performance of water tank level. The rationale of robust control and theoretical algorithm of Lyapunov stability theorem are outlined to evaluate the effectiveness of nonlinear feedback with inverse tangent function in terms of improving robustness of PID (Proportional–Integral–Derivative) controller and energy-saving capability. By demonstrating five simulations of different scenarios, it ultimately proves that the modified robust PID controller by inverse tangent function meets the requirement of energy-saving capacity. Comparing with the routine PID control, the mean control input of controlling water tank level can be reduced up to 39.2% by using modified nonlinear feedback controller. This nonlinear feedback PID controller is energy efficient and concise for its convenient use, which is feasible to expand its utility to other applications.

**Keywords:** inverse tangent function; nonlinear feedback; robust control; water level control

---

## 1. Introduction

Water tank level control is regarded as a benchmark problem and also continually studied in the domain of control systems to evaluate the control performance of a new algorithm. The application of a water tank level control is widely applied in many areas to meet the industry demands. Taking ocean vessel as an example, various tanks are fitted onboard a ship for performing different services to harmonize a safe voyage, such as oil tanks, ballast water tanks, fresh water tanks, water tanks in boilers, and even sludge tanks, etc. Taking the increment of the complexity and uncertainties of the control system into account, the nonlinear characteristics of control system are increasingly advocated in designing a controller, despite of the wide implementation of conventional PID (Proportional–Integral–Derivative) controller, which still has the advantage of conciseness, reliability, and physical significance in the industry. The nonlinear relationship between control quantity and deviation signal can be effectively mirrored by nonlinear PID control, which overcomes the drawbacks of conventional PID control to a certain extent. Especially the concept of model-based design is overwhelmingly promoted in designing a control system [1].

Reference [2] illustrated the principle and algorithm of dynamic matrix predictive control, whereas the experiments were completed for the analysis of different models for one, two, and three tanks in order to set forward a typical double tank model compared with the conventional PID model for the purpose of improving the effectiveness of the method in the control process. Moreover, a mathematical model described by two nonlinear ordinary differential equations to focus on the

adaptive control of the nonlinear system was presented in the reference [3]. Reference [4] discussed PLC-based (Programmable Logic Controller) fractional-order controller design for an industry-oriented water tank volume control application with the purpose of testifying stability and robustness properties of fractional-order discrete PID feedback-loops for different approximation methods and orders. Reference [5] claimed that the water tank control experiment would practice most knowledge of process control and introduced both linear and nonlinear tanks as examples to highlight the knowledge points of process control. The effort made in reference [6] concentrated on designing a new PID control in light of acquiring higher dynamical accuracy, which would be better applied with teaching significance. The dynamic output feedback control problem for a class of nonlinear multiple time-delays system was addressed by Zheng et al, and the flexibility and effectiveness of such a method had been verified through the simulations [7]. Reference [8] put forward a method of nonlinear control of two-tank hybrid system using sliding mode controller with a fractional-order PID sliding surface. In addition, reference [9] analyzed the tank level control problem and scrutinized a fractional order proportional integral (FOPI) controller and fractional order proportional derivative (FOPD) controller in the outer and the inner loops to compare the performance in term of energy efficiency. Guo et al., proposed an application example of the optimal control of water level oscillations in surge tank, and then summarized the optimal control of the water level oscillations in a surge tank under combined operating conditions depends on the selection of the superimposition time and the optimization of the mode of the load adjustments under the superimposition operating condition [10]. Another PI-type controller was designed by Dariusz in order to maintain a stable surge tank water level [11]. The nonlinear robust controller has also been widely used in many other fields; references [12–14] constructed a concise controller and concluded parameters for testing the strong robustness and superior control performance through the simulation test of smart autonomous surface ships. Additionally, they presented a backstepping control, which is a nonlinear controller design algorithm to design a course-keeping controller for ships, such as a nonlinear control for the steering wheel, which is also carried out in the Control System of Autonomous Vehicles. Similarly, references [15–17] researched nonlinear control, such as a robust adaptive fuzzy neural network control algorithm to design a PID controller for heading control of unmanned marine vehicles and a Non-Linear Dynamic Inversion Control to compare the primary differences between three multi-rotor platforms, and a closed-loop control system were implemented where applicable. Some other nonlinear control models were also established and proposed from different perspectives [18–21]. Be that as it may, all aforementioned references were researched thoroughly from different points of view and attained certain achievements with respects to water tank level control by using different control modes. However, from an energy-saving point of view, not much research had been undertaken, and there is still some potential that can be studied further to propose a more energy-efficient control mode.

Motivated by the above considerations, this paper explores an inverse tangent functional nonlinear feedback control and carries out a case study related to the water tank level control. Compared with the existing study, the main contributions of this paper are as follows: (i) a novel inverse tangent functional nonlinear algorithm and model are proposed and proved to design a nonlinear feedback controller for evaluating water level control; (ii) the energy saving performance is achieved by means of reducing the amplitude of control input at different flow rates.

The remaining part of this paper is organized as follows. Related theories, mathematical models, novel algorithms, and the controller design are elaborated in Section 2. Stability analysis and mathematical proof are carried out in Section 3 by using the Lyapunov stability theory, Young's inequality, and so on. The evaluation of water level control performance by simulating different scenarios is illustrated in Section 4. In the last section, the final conclusion of this study is summarized in accordance with research results acquired by the simulations, and potential work in the future is described. The water tank level control is considered to be a benchmark problem for testing new control models and algorithms. Bearing in mind the limitation and restrictions of carrying out real-world experiments, it is, however, still of keen interest and significance for us to explore actual

experiments elsewhere to further compare our research. Such motivation will stimulate our passion for research in this field continuously.

## 2. Related Theories and Mathematical Model

Unquestionably, a typical PID control possesses the features of conciseness, reliability, and obvious physical significance, which have already been widely applied in the control engineering area. The classical PID control is therefore gradually developed into many different modified modes, such as the self-tuning PID control, self-adaptive PID control, gain scheduling PID control, robustness PID control, etc., for the sake of eliminating the deficiencies that the classical PID control has shown, namely, its difficulties of integrating parameters, poor self-adaptive ability, poor robustness, low control accuracy, and so on [22,23]. The algorithm of robust PID in this paper is derived from the closed-loop gain control theories and models. The calculation process is simplified but with obvious physical significance. The robustness of the PID controller, which is designed by using this method, will be better [24–27].

### 2.1. Mathematical Model And Calculation of PID Parameters

Definition: If the coefficient of the second order strictly proper plant *G*, which is:

$$G = \frac{b_1 s + b_0}{a_2 s^2 + a_1 s + a_0}$$

If $b_1 = 0$ ($b_0, b_1, a_0, a_1, a_2$ are normal constant coefficients), then *G* is known as the second order strictly proper plant [22,23,28]. In order to eliminate one parameter, the abovementioned formula can also be simplified as:

$$G = \frac{b_0}{a_2 s^2 + a_1 s + 1}$$

The significance of the above model is to provide a way of calculating parameters of the PID controller through a closed-loop gain formation algorithm.

An actual engineering plant can be transformed strictly into the second order proper plant by reducing order of the model or approximating the Bode plotting diagram (see Figure 1).

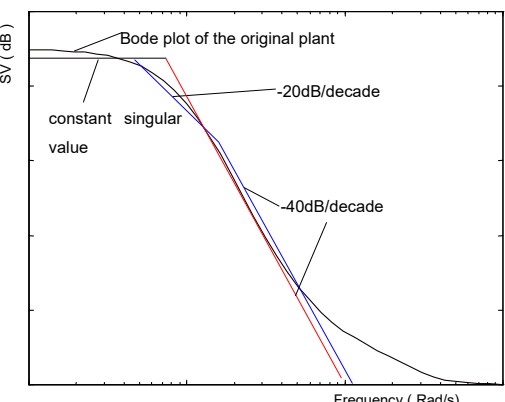

**Figure 1.** Approximation of singular value curve of the original plant by a second-order deep strictly proper plant.

By analyzing a singular value curve, we know that the actual control plants are normally low-pass and can be transformed into the second order deep strictly proper plant. It is feasible to apply in the practical engineering when such a deep strict plant is fixed with a constant and singular value at the

low frequency stage and approximated with a singular value curve of asymptote slope as -40 dB/dec or -20 dB/dec plus -40 dB/dec at high frequency stage. Then, we can get the formula as follows:

$$K = \frac{a_2 s^2 + a_1 s + a_0}{b_0 T_1 s} = \frac{a_1}{b_0 T_1} + \frac{a_0}{b_0 T_1 s} + \frac{a_2 s}{b_0 T_1} \tag{1}$$

Equation (1) is a standard PID controller, its parameters are:

$$K_p = \frac{a_1}{b_0 T_1}, K_i = \frac{a_0}{b_0 T_1}, K_d = \frac{a_2}{b_0 T_1}$$

where $1/T_1$ is the system bandwidth frequency.

### 2.2. The Application of Robust PID Control in Water Tank Level

#### 2.2.1. Mathematical Model of Water Tank Level Control

For a single tank system (see Figure 2), $Q_i$ stands for the steady-state value of input water flow, and $\Delta Q_i$ is the increment of input water flow. $Q_o$ stands for the steady-state value of output water flow, and $\Delta Q_o$ is the increment of output water flow. $h$ is the height of water level, and $h_0$ is the steady-state value of water level, $\Delta h$ is the increment of the water level. $u$ is the opening value of the adjustable input valve. $A$ is the cross sectional area, $R$ is the water resistance at output valve, and $V$ is the water volume in the tank. Based on the relationship of materials balance, under normal working conditions, the initial balance of the tank will be expressed as: $Q_o = Q_i$, $h = h_0$. When there is an increment $\Delta u$ of adjustable input valve, the actual water level will be changed accordingly. Consequently, the output water flow will be changed primarily due to the change of water level under the condition of the remaining unchanged output valve.

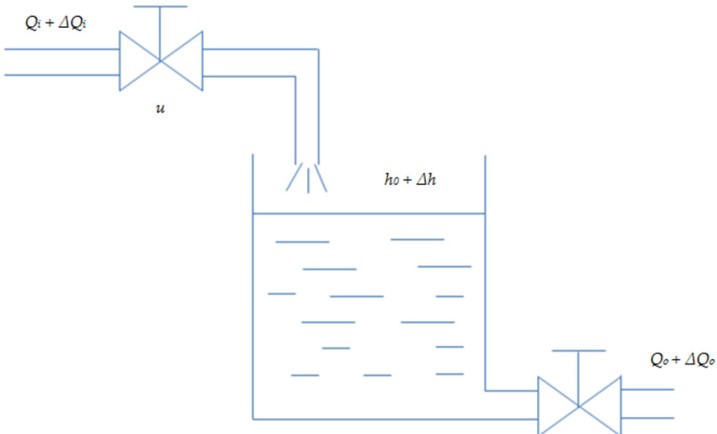

**Figure 2.** Single water tank.

The difference between the input and output water flow is:

$$\Delta Q_i - \Delta Q_o = \frac{\mathrm{d}V}{\mathrm{d}t} = A\frac{\mathrm{d}\Delta h}{\mathrm{d}t} \tag{2}$$

In formula (2), $\Delta Q_i$ is caused by $\Delta u$, then:

$$\Delta Q_i = K_u \Delta u \tag{3}$$

where $K_u$ is the constant of water flow of valve.

The relationship between output water flow and the height of the water level is:

$$Q_O = A_o \sqrt{2gh}$$

The above formula can be further linearized at balance point $(h_0, Q_o)$, then:

$$R = \frac{\Delta h}{\Delta Q_o} \tag{4}$$

We can input the formulas (3) and (4) into (2), and apply Laplace transformation, so we can get the transfer function of the single water tank:

$$G(s) = \frac{H(s)}{Q_i(s)} = \frac{K_0}{s(T_0 s + 1)} \tag{5}$$

We assume that the height of the tank is 2 m, the area of tank base is 1 m², the sectional area of the pipe is 0.05 m², initial water level is 0.5 m, and the maximum quantity of water intake is 0.5 m³/s. In accordance with the model given in the reference [29], the formula of water level and inflow volume is:

$$G(s) = \frac{H(s)}{Q_i(s)} = \frac{K_0}{s(T_0 s + 1)} = \frac{0.8}{s(2s + 1)} \tag{6}$$

where $H$ is the water level in the tank and $Q_i$ is the input flow rate.

Equation (6) can be transformed to:

$$T_0 \ddot{H} + \dot{H} = K_0 Q_i \tag{7}$$

If the disturbance term is considered, then the above model can be modified to:

$$T_0 \ddot{H} + \dot{H} = K_0 Q_i + w \tag{8}$$

where the $w$ is limited disturbance term, and also $\|w\|_\infty \leq \rho$.

### 2.2.2. Design of Water Tank Level Controller Based on the Closed-Loop Gain Algorithm

In accordance with the mathematical model of formula (6), the closed-loop gain algorithm is used to design the controller (see Figure 3). If the frequency of bandwidth is set as $1/T_1$, then the complementary sensitivity function of water tank level control is:

$$\frac{G(s)K(s)}{1 + G(s)K(s)} = \frac{1}{T_1 s + 1} \tag{9}$$

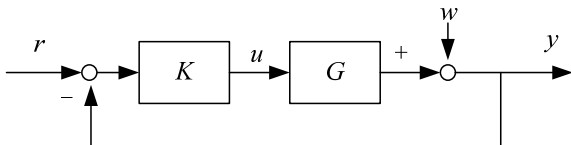

**Figure 3.** Standard feedback control system.

Substituting (6) into (9), and the cut-off frequency is deemed as $1/T_1 = 1$ rad/s, we can get all parameters of robust PID controller with:

$$K_p = \frac{1}{0.8}, K_i = 0, K_d = \frac{2}{0.8}$$

The following assumptions are made to design the water tank level controller $K$ based on closed-loop gain shaping algorithm: (1). the bandwidth frequency of the closed system is set to $1/T_1$; (2). the high frequency asymptote slope is fixed to -20dB/dec; (3). frequency spectrum of the closed-loop system is made equal to the frequency spectrum of a first-order inertial system with the largest singular value approximately one.

Then, the robust water tank level controller is:

$$K(s) = \frac{Q_i}{e} = \frac{1}{K_0 T_1} + \frac{T_0}{K_0 T_1}s \tag{10}$$

where $1/T_1 = 1\,\text{rad/s}$.

This paper focuses on both mathematical and experimental application of using nonlinear inverse tangent function which is described as:

$$u = K\arctan(we)$$

where e is the system error and e = r - y, w is the system frequency. The parameters K and w in the above function is obtained as 1/1.58 and 2, respectively, based on a number of simulations. Therefore, $\arctan[2(r-y)]/1.58$ is used to modify system error ($e$) by regulating a nonlinear feedback of a water tank level system. The new PID controller, which is modified by the inverse tangent function, will be tested for its energy-saving performance.

## 3. The Stability Analysis and Mathematical Proof of the Designed Controller

The Lyapunov stability theory is used to carry out the stability analysis of the feedback controller, which is designed based on the closed-loop gain algorithm.

Firstly, the status variable of the control system is defined as $X = \begin{bmatrix} x_1 & x_2 \end{bmatrix}^{\text{T}}$:

$$\begin{cases} x_1 = e \\ x_2 = \dot{x}_1 = -\dot{H} \end{cases} \tag{11}$$

where $e$ is the deviation between the actual and the designed water level. Then, we can get the formula of flow rate of the water tank control system based on Formula (11):

$$Q_i = Ke = \frac{1}{K_0 T_1}e + \frac{T_0}{K_0 T_1}\dot{e} \tag{12}$$

Substituting (8) into (12) to obtain:

$$T_0\ddot{H} + \dot{H} = K_0\left(\frac{1}{K_0 T_1}e + \frac{T_0}{K_0 T_1}\dot{e}\right) + w \tag{13}$$

Then by merging Formulas (11) and (13), we can get the state equation of the controlled system:

$$\dot{X} = AX + Bw \tag{14}$$

where:

$$A = \begin{pmatrix} 0 & 1 \\ -\frac{1}{T_0 T_1} & -\frac{T_1 + T_0}{T_0 T_1} \end{pmatrix}, B = \begin{pmatrix} 0 \\ -\frac{1}{T_0} \end{pmatrix}.$$

For the Equation (14), the Lyapunov function of the system is defined as:

$$V(x) = X^{\text{T}}PX \tag{15}$$

where $P$ is a positive definite real symmetric matrix, and $P = P^T$.

Disregard of the external disturbance term $w$, then:

$$\dot{V}(x) = X^{\mathrm{T}}PAX + (AX)^{\mathrm{T}}PX = X^{\mathrm{T}}(PA + A^{\mathrm{T}}P)X \tag{16}$$

According to the Lyapunov stability theory, for any defined positive definite real symmetric matrix, the following function is required in order to make sure the system is stable in the large domain at the origin:

$$A^{\mathrm{T}}P + PA = -Q \tag{17}$$

Define positive definite real symmetric matrix $Q = I = \begin{bmatrix} 1 & 0 \\ 0 & 1 \end{bmatrix}$, and make positive definite real symmetric matrix $P$ as:

$$P = \begin{pmatrix} P_{11} & P_{12} \\ P_{12} & P_{22} \end{pmatrix} \tag{18}$$

Inputting (14) and (18) into (17), to obtain:

$$\begin{pmatrix} 0 & 1 \\ -\frac{1}{T_0 T_1} & -\frac{T_1 + T_0}{T_0 T_1} \end{pmatrix} \begin{pmatrix} P_{11} & P_{12} \\ P_{12} & P_{22} \end{pmatrix} + \begin{pmatrix} P_{11} & P_{12} \\ P_{12} & P_{22} \end{pmatrix} \begin{pmatrix} 0 & 1 \\ -\frac{1}{T_0 T_1} & -\frac{T_1 + T_0}{T_0 T_1} \end{pmatrix} = -\begin{pmatrix} 1 & 0 \\ 0 & 1 \end{pmatrix} \tag{19}$$

Expanding formula (19) yields the solutions as:

$$\begin{cases} P_{11} = \frac{(T_1 + T_0)^2 + (1 + T_0 T_1)}{2(T_0 + T_1)} \\ P_{12} = \frac{T_0 T_1}{2} \\ P_{22} = \frac{T_0 T_1 (1 + T_0 T_1)}{2(T_0 + T_1)} \end{cases} \tag{20}$$

The following functions must be met to maintain the positive definiteness of matrix $P$:

$$\begin{cases} P_{11} > 0 \\ P_{11}P_{22} - P_{12}{}^2 > 0 \end{cases} \tag{21}$$

By solving the functions (21), we obtain:

$$\begin{cases} T_1 > \frac{-3T_0 + \sqrt{5T_0{}^2 - 4}}{2} \\ T_0 T_1 > 0 \end{cases} \tag{22}$$

Thus, in the case of $T_0 > \sqrt{0.8} = 0.9$, if $T_1 > 0$ is assured, then the controller will be maintained balanced state in the large domain at origin.

In order to further prove the robustness to external disturbance of the controller presented in this paper, we consider the external disturbance term $w$ in formula (9); then, the derivation of the Lyapunov function as expressed in formula (10) is derived as:

$$\dot{V}(x) = -X_1{}^2 - X_2{}^2 - T_1 w X_1 - k w X_2 \tag{23}$$
$$\text{where } k = T_1(1 + T_0 T_1)/(T_1 + T_0).$$

Based on the Young's inequality, we can obtain:

$$\begin{cases} -wTX_1 \leq T_1{}^2 X_1{}^2 + \frac{w^2}{4} \\ -kwX_2 \leq k^2 X_2{}^2 + \frac{w^2}{4} \end{cases} \tag{24}$$

Then:

$$\begin{aligned}
\dot{V}(x) &= -\left(1 - T_1^2\right)X_1^2 - \left(1 - T_1^2\right)X_2^2 + \frac{w^2}{2} \\
&\leq -\left(1 - T_1^2\right)X_1^2 - \left(1 - k^2\right)X_2^2 + \frac{\rho^2}{2}
\end{aligned} \tag{25}$$

Defining $a = \max(T_1, k)$, then:

$$\dot{V}(x) \leq -\left(1 - a^2\right)X^{\mathrm{T}}X + \frac{\rho^2}{2} \tag{26}$$

From formula (26), it can be inferred that when the state variable of the system $\|X\|_\infty > \rho/\sqrt{2(1 - a^2)}$, then $\dot{V}(x) < 0$, i.e., the state equation of the controlled system of water tank attributes uniform ultimate boundedness.

In order to carry out further discussion on the robustness for the external disturbance, $L_2$, which is the performance index of gain robust control, is defined as:

$$\int_0^t \|X\|^2 \, \mathrm{d}t \leq \mu_1 \int_0^t w^2(t)\mathrm{d}t + \mu_2 \tag{27}$$

where $\mu_1$ and $\mu_2$ are small positive numbers; then, we can obtain the following theorem.

Theorem: when designing the controller for the water tank control system as expressed by formula (14) based on the closed-loop gain algorithm, and the parameter $T_1$ of controller meeting the requirement of $0 < T_1 \leq 1$, the state variable of the system $\|X\|_\infty > \rho/\sqrt{2(1 - a^2)}$ assures the whole controller to be of uniform ultimate boundedness and also obtains the performance index of gain robust $L_2$, which is related to the controller parameter $T_1$.

To prove the above theorem, integrating formula (26) from $t = 0$ to $t = t_0$, we obtain:

$$V(t_0) + (1 - a^2)\int_0^{t_0} X^{\mathrm{T}}X\mathrm{d}t - \frac{1}{2}\int_0^{t_0} w(t)^{\mathrm{T}}w(t)\mathrm{d}t \leq V(0) \tag{28}$$

According to the definition of $V(t_0)$, formula (28) can be derived as:

$$\int_0^{t_0} \|X\|^2\mathrm{d}t \leq \frac{1}{2(1 - a^2)}\int_0^{t_0} w^2(t)\mathrm{d}t + \frac{1}{1 - a^2}\varepsilon \tag{29}$$

After valuing the range of $a$ as expressed in the formula (29), $a$ can be obtained as [30]:

$$a = \begin{cases} T_1, & T_0 \geq 1 \\ \frac{T_1(1 + T_0 T_1)}{T_1 + T_0}, & 0 < T_0 < 1 \end{cases} \tag{30}$$

The condition of $1 - a^2 > 0$ has to be assured, for the sake of maintaining the negative definite of (26); then, the value range of controller parameter can be derived based on (30):

$$0 < T_1 \leq 1 \tag{31}$$

Hence, in accordance with the result of formula (27), which is the performance index of gain robust and formula (29), when the state variable of water tank control system meets the condition of $\|X\|_\infty > \rho/\sqrt{2(1 - a^2)}$, the parameter $T_1$ of the controller of the closed-loop gain control law can be properly adjusted within the value range $0 < T_1 \leq 1$. Then, the water tank control system as expressed in formula (14) will get the performance index of gain robust $L_2$, as expressed in formula (27), and the performance index of gain robust $L_2$ is $1/2(1 - a^2)$ in relation with the parameter of controller $T_1$.

If the nonlinear inverse tangent function is used to modify system error ($e$):

$$Q_i = K \arctan(\omega e) \tag{32}$$

When $\omega e$ is not too large, $Q_i \approx K\omega e$, and the above certificate is still valid.

## 4. Simulation Experiments and Results

Simulating investigation is carried out by using Simulink toolbox, and its diagram is shown as Figure 4. For the sake of contrast, the robust PID control as presented by reference [22] had been proved and tested with better robust stability and conciseness comparing the original fuzzy PID controller as presented in reference [29]. However, the modified PID controller as presented by this paper is an inverse tangent functional nonlinear feedback control and aims to achieve energy efficient performance on the basis of the aforementioned two types of PID controllers. The main comparison is carried out between the linear PID control as presented in the reference [22] and the nonlinear PID control as proposed in this study based on the inverse tangent function.

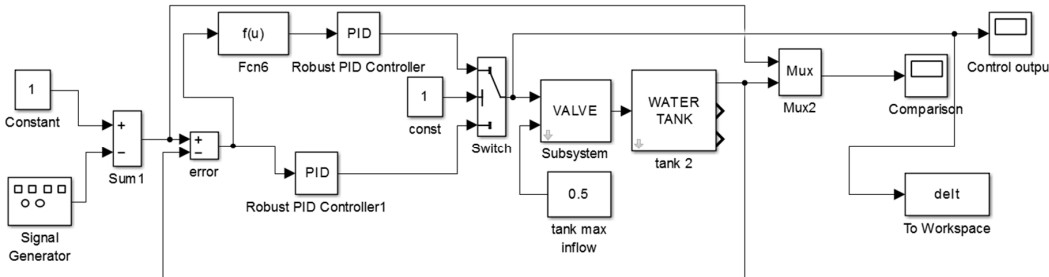

**Figure 4.** Simulation diagram of linear robust PID (Proportional–Integral–Derivative) controller and nonlinear robust PID controller modified by inverse tangent function.

The control input, which is generated by the difference between the set value and measured value, is processed by the nonlinear function, i.e., the inverse tangent function, in this study. The main impact of this inverse tangent function is to restrict the extreme signal so as to eliminate the effect of the amplitude of the control output. The detailed theoretical proof is scrutinized in the above sections.

The input signal is a square wave with an amplitude varying from 0.5 m to 1.5 m. The function (6), which is derived from reference [29], was used to design a PID controller, considering the more complicated nonlinear mathematical model used for simulation experiments as illustrated in reference [29]. When the maximum inflow rate of tank is maintained at 0.5 m³/s, the effect of nonlinear feedback, the mean control input is 0.62 m³/s, and the maximum absolute value is 176.1 m³/s. The simulation results are shown in the Figure 5a,b, where (a) is the curve of control input and (b) is the curve of the water level. There is no overshoot, and quick tracking is achieved.

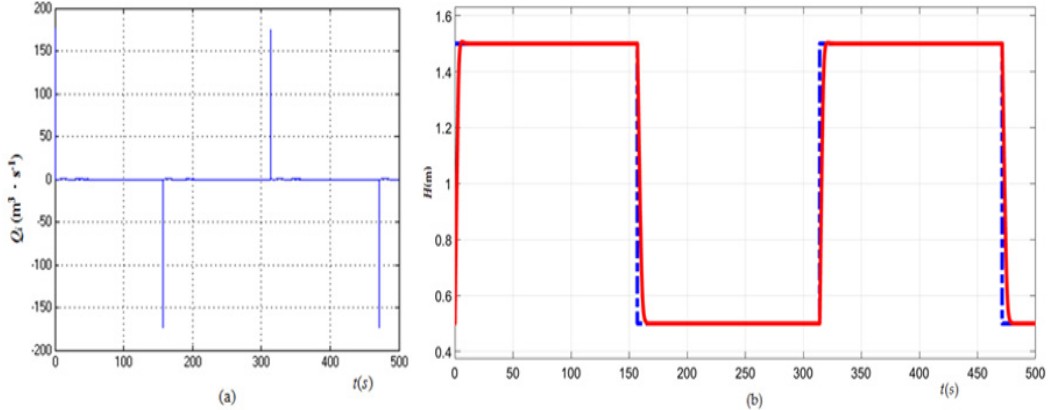

**Figure 5.** (**a**) and (**b**). Simulation results of nonlinear feedback of the modified PID controller. The inflow rate is 0.5 m³/s. (**a**) Curve of control input. (**b**) Curve of water level.

As shown in Figure 6a,b, the effect of linear feedback mean control input is 1.03 m³/s, and the maximum absolute value is 251.25 m³/s. The control input is reduced by 39.8% comparing nonlinear feedback and linear feedback. The maximum amplitude reduced by 29.9%; the goal of energy-saving by using inverse tangent function of nonlinear feedback control is therein achieved.

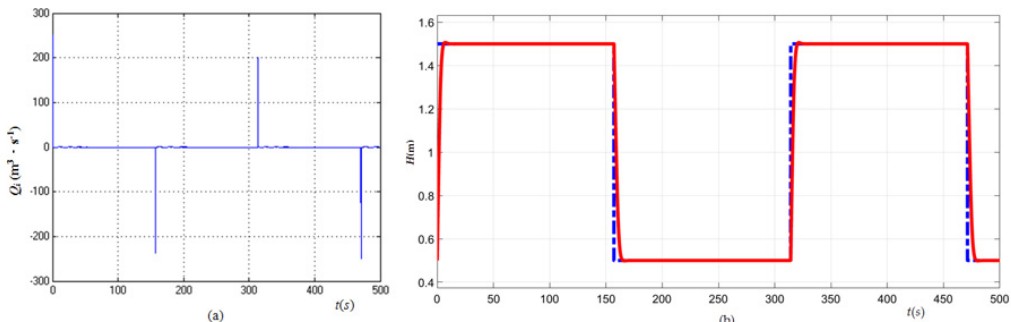

**Figure 6.** (**a**) and (**b**). Simulation results of linear feedback of the robust PID controller. The inflow rate is 0.5 m³/s. (**a**) Curve of control input. (**b**) Curve of water level.

The effect of linear feedback mean control input is 0.015m³/s. If the amplitude of input is limited between ±0.5, as shown in Figure 7a,b, the control input average of nonlinear feedback is 0.023 m³/s; the purpose of energy-saving is not achieved.

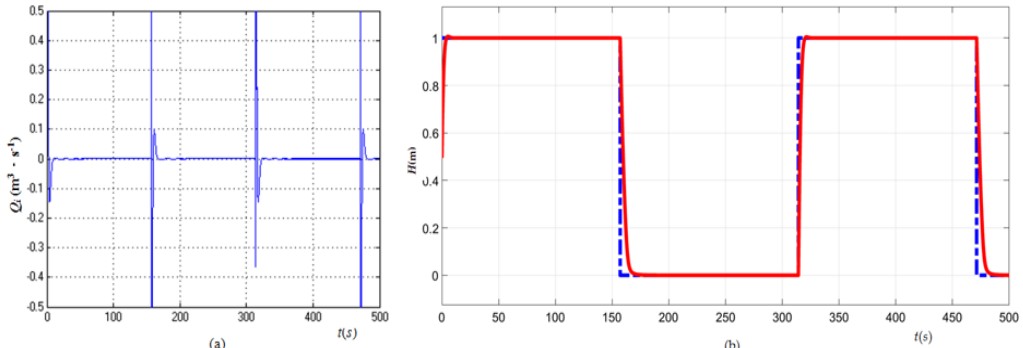

**Figure 7.** (**a**) and (**b**). Simulation results of linear feedback control with limited amplitude of input. The inflow rate is 0.5 m³/s, amplitude of input is limited. (**a**) Curve of control input. (**b**) Curve of water level.

The effect of linear feedback as shown in Figure 8a,b; mean control input is 0.88 m³/s. If the flow rate is reduced from 0.5 m³/s to 0.4 m³/s, and the limit of flow rate is set to 0.4 m³/s, the maximum absolute value is 251.3 m³/s, which is primarily due to the limit of the flow rate in the model.

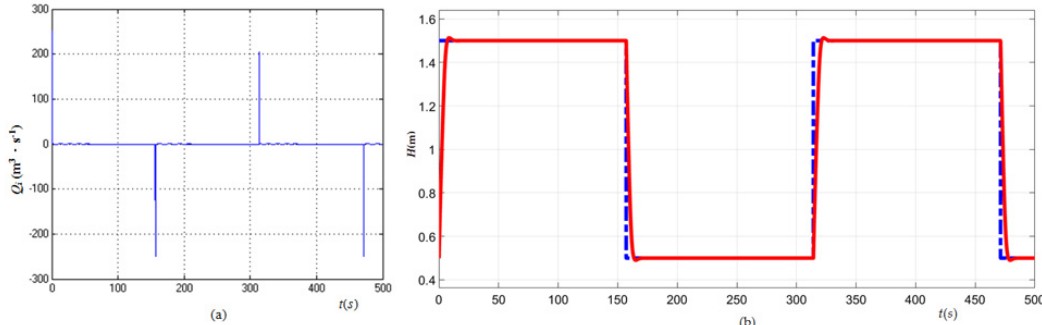

**Figure 8.** (**a**) and (**b**). Simulation results of linear feedback control with limited flow rate. The inflow rate is 0.4 m³/s. (**a**) Curve of control input. (**b**) Curve of water level.

The effect of nonlinear feedback, as shown in Figure 9a,b: the mean control input is 0.54 m³/s and the limit of flow rate is 0.4 m³/s. The maximum absolute value is 176.1 m³/s, which is primarily due to the limit of the flow rate in the model. The mean control input is reduced by 38.6% comparing nonlinear feedback and linear feedback. The purpose of energy-saving is met and the maximum amplitude is reduced by 29.9%.

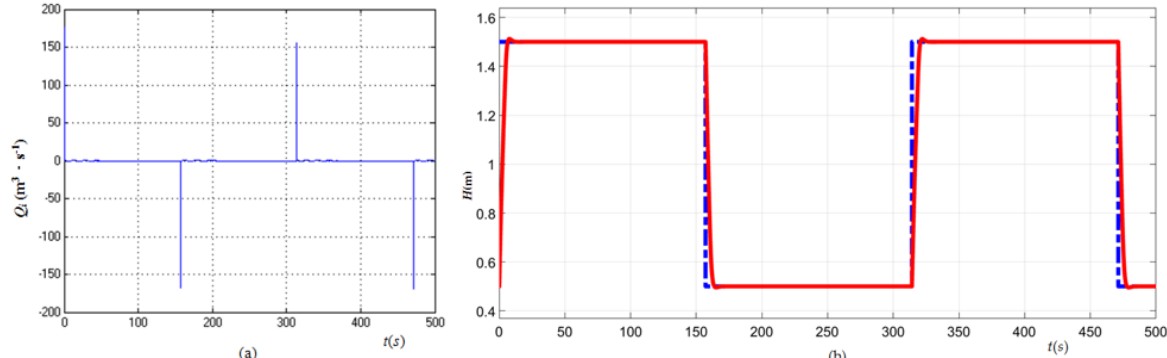

**Figure 9.** (**a**) and (**b**). Simulation results of nonlinear feedback control with limited flow rate. The inflow rate is 0.4 m³/s. (**a**) Curve of control input. (**b**) Curve of water level.

A quantitative comparison as presented in Table 1 is performed to summarize the above control simulations results, and it confirms the effectiveness of the proposed algorithm.

**Table 1.** Comparison of linear and nonlinear performances between two algorithms.

| Items | The Maximum Inflow Rate Is 0.5 m³/s | | The Maximum Inflow Rate Is 0.4 m³/s | |
|---|---|---|---|---|
| | Mean control input (m³/s) | Maximum absolute value (m³/s) | Mean control input (m³/s) | Maximum absolute value (m³/s) |
| Nonlinear feedback control | 0.62 | 176.1 | 0.54 | 176.1 |
| Linear feedback control | 1.03 | 251.25 | 0.88 | 251.3 |
| Improved performance (%) | 39.8% | 29.9% | 38.6% | 29.9% |
| Average of improved mean control input | | | 39.2% | |

## 5. Conclusions

The significant advantage of selecting nonlinear feedback is that it makes no demands on the designer beyond that of putting up for a moment with one regulating parameter, and that is what all researchers demand of their control algorithms—to be energy-efficient at no inconvenience to themselves. This paper proposes a novel modified nonlinear control based on inverse tangent function, of which the stability and feasibility is mathematically proved. Two kinds of PID controller are compared with respects of different performances in term of controlling water tank level, i.e., an inverse tangent functional nonlinear feedback PID control and a routine linear PID control, respectively. From the simulation experiments, when the flow rate is set as 0.5 m³/s, the mean control input and maximum absolute value can be improved by 39.8% and 29.9%, respectively, whereas when the flow rate is set to 0.4 m³/s, the mean control input and maximum absolute value can be improved by 38.6% and 29.9%, respectively. The control mode attributed to the nonlinear feedback control is more energy-efficient, which can reduce the average of mean control input by 39.2% in accordance with the simulation experiments. Even so, the ultimate energy-saving capability is also subject to the magnitude of input amplitude. The modified PID controller as presented in this paper is derived on the basis of the robust PID controller designed in the reference [22], whereas the characteristics of robust stability and conciseness are duly maintained but the energy-saving capacity is remarkable. This achievement

is contributed by the notion of introducing the inverse tangent function to the nonlinear feedback control, and therefore, the expected objective of using less energy is attained. However, there are still some conditions and limitations to carry out real experiments, which will be addressed in the future research, especially for the application of the liquid level control onboard a vessel. The modification method analyzed in this paper for a robust PID controller by applying inverse tangent functional nonlinear feedback is tested for the merits of being energy efficient and concise for its convenient use, and could be applied to a larger extent for other areas to meet the industry demands.

**Author Contributions:** Visualization, J.Z.; Resources, J.Z., Validation, J.Z.; Writing—Review and Editing, J.Z.; Methodology, X.Z.; Supervision, X.Z. All authors have read and agreed to the published version of the manuscript.

**Funding:** This work is partially supported by the National Science Foundation of China (Grant No.51679024), the Fundamental Research Funds for the Central University (Grant No.3132016315), and the University 111 Project of China (Grant No.B08046).

**Conflicts of Interest:** The authors declare no conflict of interests.

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
