# Peer review of "Inverse Tangent Functional Nonlinear Feedback Control and Its Application to Water Tank Level Control"

_processes, doi:10.3390/pr8030347_

Round 1
Reviewer 1 Report
The manuscript deals with a design and experimentation of modified robust PID controller using the inverse tangent function. Authors stipulate the energy-saving ability of the modified controller. The paper is structurally written and self-explainable. The idea is supported by Lyapunov stability theory.
I have some minor and a major comment. In the experimental part, authors have shown only the simulations. I am convinced that for publication, authors should design a real-world experiment as well, to show practical utility of the proposed idea. Especially, because they propose a possible application of shipboard tank level, i.e. "IMO".
Some other minor comments:
- Organization of results part should be improved in order to attract more readers, i.e. currently it is difficult to compare the actual results between the usual and modified PID. The figures are more or less sequentially scattered.
- Figures should be provided in a more clear format, e.g. pdf, eps. Reference values (blue lines) might be visualized using the dashed line in order to improve the clarity of figures and expose the actual signal (green lines).
- The results lines (Figs. 5-9) should be of higher thickness. Currently these are heavily visible if you print the manuscript on paper.
Author Response
Dear Editor and reviewer,
Many thanks for your supervision of the reviewing process of our manuscript ID “Processes-725324” and titled with “Inverse Tangent Functional Nonlinear Feedback Control and Its Application to Water Tank Level Control”. Thanks for the chance of revision you gave us and we sincerely appreciate your efforts in censoring this paper before and after. My supervisor, I and my colleagues in our team worked on this study again and revised this manuscript in accordance with reviewer’s comments. It is exactly because of the academic justice and righteousness that we want to publish our work in this prestigious journal.
We also want to express our sincere thanks to the reviewers. It is exactly because of their keen suggestions and generous sharing of their knowledge that promotes a significant enhancement of this paper. We have studied each comment of the reviewers carefully and tried our best to make responses in a sound way. A major revision has been conducted in the revised version, and the changing places were marked in red. Corresponding to each comment, we have made the response as following.
|
Review’s comment 1: I have some minor and a major comment. In the experimental part, authors have shown only the simulations. I am convinced that for publication, authors should design a real-world experiment as well, to show practical utility of the proposed idea. Especially, because they propose a possible application of shipboard tank level, i.e. "IMO". |
|
Author response: Thank you very much for the suggestions. We also agreed with review’s point. However, this is a novel concept of formulating a nonlinear feedback control based on inverse tangent function. Considering the fact that water tank level control has been regarded as a benchmark problem, we therefore started to design a nonlinear controller and carry out associated simulation tests to check the feasibility of our research. The means of using mathematical derivation and simulation is also considered as effective methods to test the benchmark problem, especially when the real-world experiments could not be carried out with some limitations or restrictions. However, we have addressed this point in our study and will implement in the future work whenever possible. |
|
Author actions: The limitations of carrying out real-world experiments were addressed in both introduction and conclusion parts as follows: “The water tank level control is considered as a benchmark problem for testifying new control models and algorithms. Bearing in mind the limitation and restrictions of carrying out the real-world experiments, it is however still of keen interest and significance for us to explore actual experiments elsewhere to further compare our research.”
“However, there are still some conditions and limitations to carry out real experiments which will be addressed in the future research, especially for application of the liquid level control onboard a vessel.” |
|
Review’s comment 2: Organization of results part should be improved in order to attract more readers, i.e. currently it is difficult to compare the actual results between the usual and modified PID. The figures are more or less sequentially scattered. |
|
Author response: Thanks a lot for this comment. We have rewritten the results part in the conclusion. |
|
Author actions: The conclusion part was reorganized as follows: “The significant advantage of selecting nonlinear feedback is that it makes no demands on the designer beyond that of putting up for a moment with one parameter regulating, and that is what all researchers demand of their control algorithms—to be energy-efficient at no inconvenience to themselves. This paper proposes a novel modified nonlinear control based on inverse tangent function, of which the stability and feasibility is mathematically proved. Two kinds of PID controller are compared with respects of different performances in term of controlling water tank level, i.e., inverse tangent functional nonlinear feedback PID control and routine linear PID control respectively. From the simulation experiments, when the flow rate is set as 0.5 m3/s, the mean control input and maximum absolute value can be improved by 39.8% and 29.9% respectively, whereas when the flow rate is set as 0.4 m3/s, the mean control input and maximum absolute value can be improved by 38.6% and 29.9% respectively. The control mode attributed to the nonlinear feedback control is more energy-efficient, which can reduce the average of mean control input by 39.2% in accordance with the simulation experiments. Even so, the ultimate energy-saving capability is also subject to the magnitude of input amplitude. The modified PID controller as presented in this paper is derived on the basis of the robust PID controller designed in the reference [22], whereas the characteristics of robust stability and conciseness are duly maintained but the energy-saving capacity is remarkable. This achievement is contributed by the notion of introducing the inverse tangent function to nonlinear feedback control and therefore the expected objective of using less energy is attained. However, there are still some conditions and limitations to carry out real experiments which will be addressed in the future research, especially for application of the liquid level control onboard a vessel. The modification method analyzed in this paper for a robust PID controller by applying inverse tangent functional nonlinear feedback is testified for the merits of being energy efficient and concise for its convenient use, and could be applied to a larger extent for other fields.” |
|
Review’s comment 3: Figures should be provided in a clearer format, e.g. pdf, eps. Reference values (blue lines) might be visualized using the dashed line in order to improve the clarity of figures and expose the actual signal (green lines). |
|
Author response: Thank you very much for this comment. We re-formatted the figures. Reference values (blue lines) were transformed into dash line and actual signal was transformed into red line to be in clearer format. |
|
Author actions: We have re-formatted the figures, taking the following figure as an example:
|
|
Review’s comment 4: The results lines (Figs. 5-9) should be of higher thickness. Currently these are heavily visible if you print the manuscript on paper. |
|
Author response: Thank you very much for the suggestion. The figures (5-9) for the comparisons were re-formatted with higher thickness. |
|
Author actions: The figures were re-formatted, please also refer the above example. All figures (5-9) in the paper were changed accordingly. |
In addition, we also have reexamined the whole paper for the language checking in terms of grammars and spellings. All changes were marked with red by using “track changes” function of the MS word. We also provide clean copy for both editor and reviewers to check our revisions.
We really grateful for comments from reviewers and editing work from editors. We will definitely improve our research ability with the full support from you.
Sincerely
Jian Zhao

Reviewer 2 Report
General remarks:
1. This is another paper that gives some modification of existed methods with application to the well-known system which has been well studied around the world and many papers have been published. I recommend to the Authors to write down more about novelty and contribution of the paper that should be clearly stated. How is the present paper different from these previous results?
2. Comparison of the inverse tangential – based nonlinear control with PID control that is a linear one, is unjustified and may not reasonable.
3. The second paragraph in the introduction gives some basic information that can be found in the textbook. I recommend to skip or concise it.
4. And the introduction is lack of paragraph with the authors' claims and paper organization.
5. Also, the two paragraphs at the beginning of section 2 give fundamental information about well-known PID controllers and should be consistent.
6. Line 130, some different font sizes is used for parameters definitions.
7. Model (6) should be validated with experimental modelling/identification.
8. How the function given inline 237 was derived? The explanation should be given.
9. Simulation tests (Fig. 5 to 9) give not reasonable outputs, e.g. level (H) is changing as step form that is not possible in reality, and also Q?
10. Section 4 does not give experimental verification as was stated.
11. Regarding results, the control input (controller effort/output) should be also analyzed.
12. The conclusions should be supported with results.
Author Response
Dear Editor and reviewer,
Many thanks for your supervision of the reviewing process of our manuscript ID “Processes-725324” and titled with “Inverse Tangent Functional Nonlinear Feedback Control and Its Application to Water Tank Level Control”. Thanks for the chance of revision you gave us and we sincerely appreciate your efforts in censoring this paper before and after. My supervisor, I and my colleagues in our team worked on this study again and revised this manuscript in accordance with reviewer’s comments. It is exactly because of the academic justice and righteousness that we want to publish our work in this prestigious journal.
We also want to express our sincere thanks to the reviewers. It is exactly because of their keen suggestions and generous sharing of their knowledge that promotes a significant enhancement of this paper. We have studied each comment of the reviewers carefully and tried our best to make responses in a sound way. A major revision has been conducted in the revised version, and the changing places were marked in red. Corresponding to each comment, we have made the response as following.
|
Review’s comment 1: This is another paper that gives some modification of existed methods with application to the well-known system which has been well studied around the world and many papers have been published. I recommend to the Authors to write down more about novelty and contribution of the paper that should be clearly stated. How is the present paper different from these previous results? |
|
Author response: Thank you very much for the suggestions. We have written down more about the novelty and contributions in the end of introduction section. |
|
Author actions: The additional part we added with respect of this comment is (marked in red in the revised version of “track changes”: “Motivated by the above considerations, this paper explores an inverse tangent functional nonlinear feedback control and carries out a case study related to water tank level control. Comparing with the existing study, the main contributions of this paper are as follows: (i) a novel inverse tangent functional nonlinear algorithm and models are proposed and proved to design a nonlinear feedback controller for evaluating water level control; (ii) the energy saving performance is achieved by means of reducing the amplitude of control input at different flow rates.” |
|
Review’s comment 2: Comparison of the inverse tangential – based nonlinear control with PID control that is a linear one, is unjustified and may not reasonable. |
|
Author response: Thank you very much for the comment. Our research is mainly based on the references of linear PID control (reference [22,23,28]), and also conventional fuzzy control as presented in reference [29]. The main comparison is carried out between the linear PID control as presented in the reference [22] and the nonlinear PID control as proposed by this paper based on inverse tangent function. |
|
Author actions: We have written more in the sections mathematical model, simulation and conclusion with respect of clarifications of the referred linear PID control and the highlights of nonlinear control modified by inverse tangent function. |
|
Review’s comment t 3: The second paragraph in the introduction gives some basic information that can be found in the textbook. I recommend to skip or concise it. |
|
Author response: Thank you for this suggestion. Yes, this part was summarized from textbook. We have deleted this part to make it more concise. |
|
Author actions: The second paragraph was deleted. |
|
Review’s comment 4: And the introduction is lack of paragraph with the authors' claims and paper organization. |
|
Author response: Thanks for this comment. Sorry for omitting the paper organization part. We have rewritten the introduction section and added more information about this comment. |
|
Author actions: We have rewritten the introduction part and added more information about the author’s claims and paper organization as follows: “The remaining part of this paper is organized as follows. Related theories, mathematical models, novel algorithms, and the controller design are elaborated in section 2. Stability analysis and mathematical proof are carried out in section 3 by using Lyapunov stability theory, Young’s inequality and so on. The evaluation of water level control performance by simulating different scenarios is illustrated in section 4. In the last section, the final conclusion of this study is summarized in accordance with research results acquired by the simulations, and also potential work in the future is anticipated. The water tank level control is considered as a benchmark problem for testifying new control models and algorithms. Bearing in mind the limitation and restrictions of carrying out the real-world experiments, it is however still of keen interest and significance for us to explore actual experiments elsewhere to further compare our research. Such motivation will stimulate our passions research in this field continuously.” |
|
Review’s comment 5: Also, the two paragraphs at the beginning of section 2 give fundamental information about well-known PID controllers and should be consistent. |
|
Author response: Thanks for this comment. Yes, this part is a little bit wordy. We have deleted some sentences and rephrased some sentences to make it consistent. |
|
Author actions: We have changed this part as follows: “Unquestionably, a typical PID control possesses the features of conciseness, reliability and obvious physical significance which have already been widely applied in the control engineering area. The classical PID control is therefore gradually developed into many different modified modes, such as self-tuning PID control, self-adaptive PID control, gain scheduling PID control, robustness PID control, etc., for the sake of eliminating the deficiencies which classical PID control has been manifested for its difficulties of integrating parameters, poor self-adaptive ability, poor robustness and the low control accuracy and so on [22-23]. The algorithm of robust PID in this paper is derived from the closed-loop gain control theories and models. The calculation process is simplified but with obvious physical significance. The robustness of PID controller which is designed by using this method will be better [24-27]. |
|
Review’s comment 6: Line 130, some different font sizes is used for parameters definitions. |
|
Author response: Thank you very much. Sorry we used the different font size for the parameters definitions. |
|
Author actions: We have changed the font size for this part and also throughout the whole paper to make all fonts in the same equivalent size. |
|
Review’s comment 7: Model (6) should be validated with experimental modeling/identification. |
|
Author response: Thanks for this comment. This Model was actually referred from the reference [29], and validation is given in this reference. For the sake of conciseness of the paper and also considering this model is not the contribution of this paper, we skipped this part but we added the reference of this model. |
|
Author actions: We have added the reference of this model, which outlines the validation of this model. |
|
Review’s comment 8: How the function given inline 237 was derived? The explanation should be given. |
|
Author response: Thank you very much. Sorry we did not provide the explanation of this function. We have reorganized this part and provided more explanation about this function. |
|
Author actions: The explanation about this function we added as follows” “This paper focuses on both mathematical and experimental application of using nonlinear inverse tangent function which is described as
where e is the system error and e = r - y, w is the system frequency. The parameters K and w in the above function is obtained as 1/1.58 and 2 respectively based on a number of simulations. Therefore, is used to modify system error (e) by regulating a nonlinear feedback of a water tank level system. The new PID controller which is modified by inverse tangent function will be testified for its energy-saving performance.” |
|
Review’s comment 9: Simulation tests (Fig. 5 to 9) give not reasonable outputs, e.g. level (H) is changing as step form that is not possible in reality, and also Q? |
|
Author response: Thank you for this comment. Yes, the simulation tests were carried out under some hypothesis and conditions in order to achieve more comparable analysis about the main focus of this study. It is a bit realistic, however we believe it will response more quickly in the real experiments. |
|
Author actions: We very much appreciate the reviewer’s comment, we will definitely take into account for our future research. |
|
Review’s comment 10: Section 4 does not give experimental verification as was stated. |
||||||||||||||||||||||||||||||
|
Author response: Thanks the comment. We have summarized the experimental verification and added a table in this part. |
||||||||||||||||||||||||||||||
|
Author actions: We have summarized the experimental verification and added a table as follows: “A quantitative comparison as presented in Table 1 is performed to summarize the above control simulations results, and it confirms the effectiveness of the proposed algorithm. Table 1. Comparison of linear and nonlinear performances between two algorithms.
|
||||||||||||||||||||||||||||||
|
Review’s comment 11: Regarding results, the control input (controller effort/output) should be also analyzed. |
|
Author response: Thank you very much for this comment. We have analyzed this point and added in section 4. |
|
Author actions: We added more analysis as follows: “The control input which is generated by the difference between set value and measured value is processed by the nonlinear function, i.e., inverse tangent function in this study. The main impact of this inverse tangent function is to restrict the extreme signal so as to eliminate the effect of amplitude of control output. The detailed theoretical proof is scrutinized in the above sections.” |
|
Review’s comment 12: The conclusions should be supported with results. |
|
Author response: Thank you. We have rewritten the part of conclusion and added more supports |
|
Author actions: We added more results as follows: “From the simulation experiments, when the flow rate is set as 0.5 m3/s, the mean control input and maximum absolute value can be improved by 39.8% and 29.9% respectively, whereas when the flow rate is set as 0.4 m3/s, the mean control input and maximum absolute value can be improved by 38.6% and 29.9% respectively. The control mode attributed to the nonlinear feedback control is more energy-efficient, which can reduce the average of mean control input by 39.2% in accordance with the simulation experiments.” |
In addition, we also have reexamined the whole paper for the language checking in terms of grammars and spellings. All changes were marked with red by using “track changes” function of the MS word. We also provide clean copy for both editor and reviewers to check our revisions.
We really grateful for comments from reviewers and editing work from editors. We will definitely improve our research ability with the full support from you.
Sincerely
Jian Zhao

Round 2
Reviewer 1 Report
My opinion is that the paper can be accepted in present form.
Authors have provided a clear revision.
Reviewer 2 Report
I have no further comments